# Decreasing the Impact of Anxiety on Cancer Prevention through Online Intervention

**DOI:** 10.3390/ijerph17030985

**Published:** 2020-02-05

**Authors:** Maksymilian Gajda, Małgorzata Kowalska

**Affiliations:** Department of Epidemiology, School of Medicine in Katowice, Medical University of Silesia, 40-055 Katowice, Poland

**Keywords:** anxiety, cancer, prevention, online, intervention

## Abstract

Background: Low levels of public knowledge, incorrect beliefs, and anxiety are the most often mentioned factors that may negatively affect the implementation of preventive campaigns and timely diagnosis of cancer. Cancer is a major unresolved problem for global public health. As a result, many effective preventive measures need to be found and implemented. Methods: For a duration of 18 months, readers of the Polish scientific Internet portal were invited to participate in the Polish On-line Randomized Intervention aimed at Neoplasm Avoidance (PORINA) study. Level of cancer-related anxiety was our main measure (self-declared on a simple five-point Likert scale) in this analysis. Results: A total of 463 participants were qualified for the final analysis. Respondents with a positive family history of cancer (*p* < 0.001) declared the highest level of cancer-related anxiety, whereas lower levels were declared by those previously treated for cancer (*p* = 0.006). The conducted educational intervention reduced the declared level of cancer-related anxiety. Conclusions: The results of this study provide evidence that the use of web-based interventions aimed at increasing awareness could reduce cancer-related anxiety and may lead to more frequent consent to undergo some of the medical procedures used to diagnose or treat cancer.

## 1. Introduction

Many factors negatively impact the implementation of preventive health campaigns. These factors also contribute to a delay in the scheduling of patient visits to the doctor. Late diagnosis and subsequently delayed treatment often lead to worse prognoses. The insufficient level of society’s knowledge, current beliefs, anxiety, embarrassment, and cultural factors may contribute to this situation [1,2,3,4,5,6,7,8,9,10,11,12,13]. So far, many actions have been taken to reduce the widespread misconceptions about cancer and associated stigmatization. Such activities have been added, among others, to the World Cancer Declaration of 2013 (Union for International Cancer Control) and the World Cancer Declaration of 2011 (goal no. 5) [14]. Unfortunately, cancer is still a significant and unresolved global public health problem, and more effective preventive procedures need to be found. At the same time, Internet resources are becoming one of the primary sources of knowledge about cancer [15,16,17,18]. Web-based features are increasingly being used by public health researchers [19,20,21]. In 2003, a social support program conducted via the Internet effectively reduced the level of depression, stress, and anxiety associated with cancer treatment [22]. Embarrassment and fear are included among the factors that negatively affect the tendency of men to search for medical information [23,24]. Many studies conducted so far involving both patients diagnosed with cancer and cancer survivors investigated the effectiveness of interventions in reducing the level of disease-related anxiety [25,26,27,28]. To the best of our knowledge, whether this type of activity is also possible in the non-patient population of Central Europe has not yet been established.

## 2. Materials and Methods

Here, we present the most important information necessary for assessing the relationship between the level of knowledge and anxiety associated with cancer. A more comprehensive description of the protocol of the Polish On-line Randomized Intervention aimed at Neoplasm Avoidance (PORINA) study was included in our previous papers on the PORINA study [29,30] as well as in a doctoral thesis (available upon request).

### 2.1. General Description of the Study

From 14 May 2015 to 13 November 2016, all readers of the publicly available Polish scientific Internet portal were invited to participate in the Polish On-line Randomized Intervention aimed at Neoplasm Avoidance (PORINA) study. A dedicated internet platform was created to allow the study to be conducted, including obtaining electronic informed consent. Before granting voluntary consent, visitors were provided all information about the study including its anonymity, aims, and stages. Participants were randomly assigned (1:1) to two groups: control or interventional. All subjects were asked to complete our questionnaire two times (baseline and final assessment). Through the evaluation of questionnaires obtained from participants in the control group, we were able to successfully validate the tool [29]. Subjects assigned to the intervention group were provided access to educational material and were required to participate in a simple quiz. This process allowed us to verify whether they were acquainted with the educational content. Failure to complete the quiz held participants back from the next phase of the study. In this case, the subjects were asked to re-enter the educational module [29,30].

### 2.2. Data Assessment and Main Measures

Our questionnaire contained only closed-ended questions written in Polish. This allowed us to assess demographic data, subjects’ general knowledge on cancer, and their attitude toward undergoing selected medical procedures applicable to the diagnosis and treatment of cancer, and evaluate the source of information on cancer. Participants were also asked to self-declare their anxiety associated with selected disease entities (including cancer). For this purpose, a simple 5-point Likert scale was used where the value 1 indicated the lowest level of anxiety and 5 indicated the highest level. The level of anxiety was our main measure in the current analysis. As described in our previous publications, the original Cancer Knowledge Index (CKI) was used to evaluate participants’ cancer-related knowledge. It was calculated based on the answers to a set of 20 questions (value range from 0 to 20). Three levels of CKI were distinguished based on tertile distribution [29,30].

### 2.3. Statistical Analysis and Ethical Approval

All statistical analyses were conducted with R software [31] and *p*-values less than 0.05 were considered statistically significant. We analyzed simple descriptive statistics, difference tests, and logistic regression models in which the level of anxiety was a dependent value and independent values included the participants’ age, sex, place of residence, occupation, level of education, family history of cancer, personal history of cancer, being treated for cancer, self-assessment of cancer-related level of knowledge, and self-declaration of willingness to improve the level of cancer-related knowledge. Given the small size of some subgroups, for the purposes of statistical analysis, the values of some qualitative variables were recoded (simplified) [30]. The protocol for this study was approved by the Bioethical Committee of the Medical University of Silesia in Katowice (KNW/0022/KB1/146/14; 24 December, 2014, Katowice, Poland).

## 3. Results

There were initially 1118 volunteers in the study, of which 463 complete responses to the questionnaire qualified for the final analysis. The final number of participants was influenced by the dropout rate of 58.6% (more than half of the participants did not participate in all stages of the study). Detailed data were presented in our previous publication [29]. Both groups were comparable in terms of demographic characteristics and oncological history [29,30]. At the beginning of the study, we initially assessed the level of anxiety in both groups. No statistically significant differences were found in the levels of anxiety declared in the initial survey by the members of the control and intervention groups (*p* = 0.28, χ^2^ test).

Among the examples of diseases listed in the questionnaire, the highest level of anxiety (score of five on a five-point Likert scale) was most often declared for the possibility of developing cancer at 56% and 59% in the control and intervention groups, respectively. The highest level of anxiety was most rarely declared in the case of influenza, with only 4.3% and 0.4% of respondents in the control and intervention group, respectively (full report available in Appendix A).

### 3.1. Does the Level of Anxiety Depend on Knowledge about Cancer?

CKI values in the initial survey were not associated with the declared level of anxiety related to developing cancer (*p* = 0.72 in Cochran–Mantel–Haenszel test; full report is available in Table 1). Similarly, differences in levels of anxiety caused by other diseases listed in the questionnaire were statistically insignificant.

### 3.2. Other Factors that May Affect the Level of Anxiety Associated with Cancer

Participants with a positive family history of cancer significantly more often (*p* < 0.001) declared a higher level of anxiety associated with this group of diseases. The percentage of people who declared the maximum level of anxiety against cancer in the first group was 61.2%, and 47.5% in the control group (negative family history of cancer).

In contrast, a significantly lower (*p* = 0.017) level of anxiety was declared by participants who had underwent oncological treatment (52.1% with score of 5 and 8.3% with score of 4) compared to participants who did not (58.3% with score of 5 and 21.7% with scores of 4). A similar association (*p* = 0.02) was observed in the group of people diagnosed with cancer who declared less severe anxiety (50% score of 5 and 14% score of 4) than healthy people (58.6% scoring 5 and 21.1% scoring 4). We did not find any other statistically significant differences in the self-assessment of anxiety associated with the possibility of developing cancer in the analyzed subgroups defined by selected demographic variables. The results are presented in Table 2 and Figure 1.

We additionally performed analysis with a logistic regression model that included the same variables as the descriptive statistics presented in Table 2. We found a positive family history of cancer to be the only significant factor of the level of anxiety with an odds ratio of 4.18 (95% confidence interval from 1.69 to 10.76, *p* = 0.002).

### 3.3. Impact of Educational Intervention on Level of Anxiety Associated with Cancer

The conducted educational intervention reduced the declared level of anxiety associated with the possibility of developing cancer (Table 3, Figure 2).

The statistical analysis (Wilcoxon test for paired variables) showed that differences in anxiety levels before and after the intervention were statistically significant (*p* = 0.03). At the same time, no statistically significant differences were found in the control group (*p* = 0.6). Additionally, an ordinal regression model analysis (CLMM function of R software) confirmed these results. There were also no statistically significant differences in the intensity of anxiety associated with other disease entities included in the questionnaire. Moreover, none of the variables was statistically significantly associated with a change in the level of anxiety. A complete listing of all disease entities included in the questionnaire is provided in Appendix A.

## 4. Discussion

Our previous analyses demonstrated that a higher level of education, age, medical occupation, and history of cancer may affect the decision to undergo necessary medical procedures [29,30]. Anxiety and embarrassment, along with lack of knowledge, misconceptions, and financial aspects are the main barriers to accessing cancer prevention [32,33,34,35]. Metwally et al. analyzed the entries published on Twitter to assess the views and attitudes of users of this social medium on screening for cancer. Fear and pain were common among negative tweets [36]. The search for the information can be an expression of a high level of anxiety, as was shown for people seeking information on human papillomavirus [37]. For example, about 66% of Polish women indicated fear of cancer as a reason for not undergoing mammography [33]. In this context, the possibility of using web-based decision aids to support decisions about screening mammography is promising [38].

The subjective nature of the assessment and the multifactorial determinants of anxiety can affect the final interpretation of the results and complicate comparisons. An additional complication in a direct comparison of the obtained results is the different methods of assessing the level of declared anxiety. In our study, a simple Likert scale, which was limited by the volume of the questionnaire, was used instead of an extensive validated psychometric tool. Ventura et al. conducted targeted interventions for reducing anxiety and depression in a group of 105 patients diagnosed with breast cancer. The study failed to achieve statistically significant improvement in the above outcomes [39]. The authors explained the lack of effects through methodological issues.

According to the results of the PORINA study presented here, the highest declared level of anxiety compared to others included in the questionnaire was induced by the possibility of developing cancer. Anxiety related to cancer has also become the subject of the Health Information National Trends Survey (HINTS) program that has been operating in the USA since 2003. It aims to obtain data about the knowledge and approach of the population to health, including cancer. Representative data on a national scale are obtained periodically by mail or telephone depending on the edition and are made available (https://hints.cancer.gov/) to researchers for further analysis [40]. As part of the fourth cycle of HINTS from 2014, the respondents were asked to assess their fears of developing cancer on a five-point scale. Every 10th respondent declared moderate (4/5) and significant (5/5) anxiety levels at 13.2% and 8.8%, respectively [40]. Therefore, these values are lower than those obtained in the PORINA study; nevertheless, a different population and slightly different questions in both studies should be noted.

Another important issue is the impact of the level of knowledge on anxiety. In the present study, we were not able to demonstrate that a low level of knowledge may be associated with a higher level of anxiety. As observed by Robb et al., an increase in knowledge did not lead to a significant change in the level of anxiety [25], in contrast to the results of our own research of a significant reduction in the intensity of declared anxiety after the educational intervention. Other authors pointed out that the level of cancer risk is an independent factor determining the increase in intensity of declared anxiety as a result of educational interventions [41]. Given the inconsistent results, it remains an open question whether the improvement of knowledge affects the level of anxiety. According to Bowen et al., the better the perception of risk factors and the higher the level of anxiety associated with cancer, the higher the effectiveness of interventions [42]. No such relationship was confirmed in our study; however, it seems appropriate to consider it in future studies.

### Advantages and Disadvantages

In our opinion, the fact that Internet users are younger is an advantage of conducting interventions using this medium. In such a population, it should be easier to create the right attitudes, including those related to the state of health and prevention of cancer. Looking at the median age of the population in the PORINA study (33 years old), this should be considered one of the strengths of our study.

However, among the weaknesses of the study is the above-mentioned measurement of anxiety by self-declared answers on a simple Likert scale. Only one method of providing educational information (the Internet) was evaluated. Still, other researchers have previously shown the equivalence of interventions using video, text, and the Internet [43]. Given the low cost of conducting Internet interventions (and the aforementioned age of the target population), this is not a significant weakness of the study.

The main reason for the occurrence of a significant difference in the number of participants originally included in the study and the group qualified for the final analysis was the participants’ resignation from further participation in the study. This is known as the dropout phenomenon (ratio of the number of people who completed the study to all enrolled), often resulting in lower response ratios and selection bias, which could have affect the results obtained. The dropout rate in our study (58.6%) seems to be consistent with other authors’ reports [44]. This topic was discussed in more detail in our first paper on the PORINA study [29].

## 5. Conclusions

The results of the study provide evidence that the use of Internet interventions aimed at increasing awareness could reduce anxiety levels associated with the possibility of developing cancer. The obtained results are promising for improving the frequency of participation in future medical procedures used to diagnose or treat cancer. However, it is necessary to verify the results in practice through a well-designed prospective study.

## Figures and Tables

**Figure 1 ijerph-17-00985-f001:**
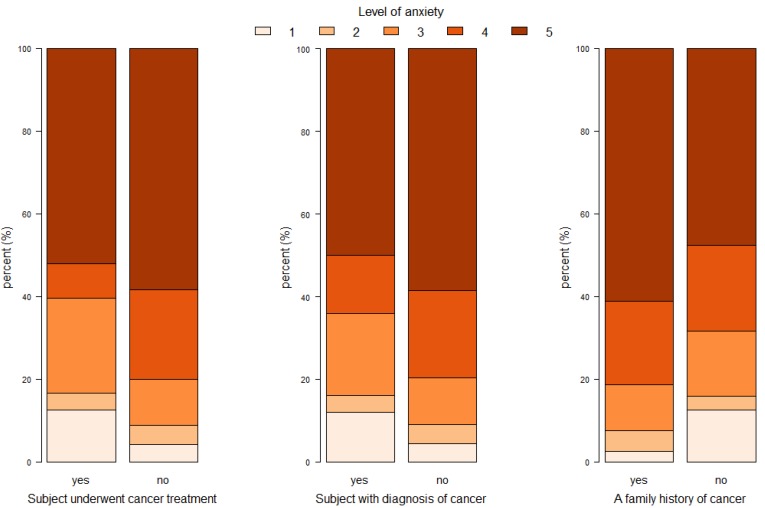
The level of anxiety depending on the declared burden of the individual and family history of cancer, as well as whether the subject had ever undergone oncological treatment.

**Figure 2 ijerph-17-00985-f002:**
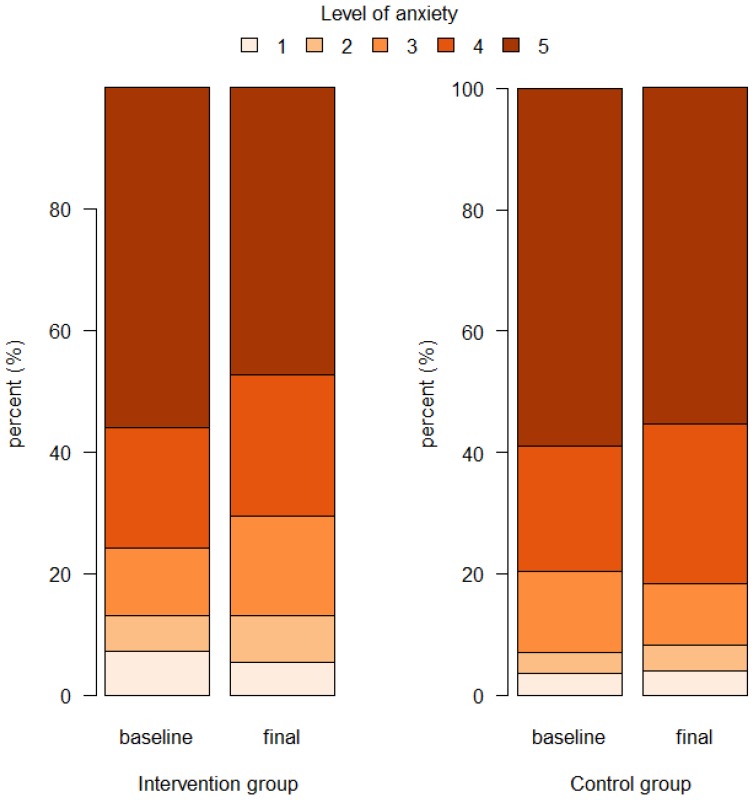
Self-declared baseline and final level of anxiety associated with cancer in intervention and control groups.

**Table 1 ijerph-17-00985-t001:** Descriptive statistics of the variable assessing differences in the severity of cancer-related anxiety in the subgroups defined by the tertile distribution of Cancer Knowledge Index (CKI) value.

Level of Cancer-Related Anxiety *		CKI Value (Baseline Survey)			
	Overall	Low	Medium	High	*p* **
	*N* = 463	*N* = 142	*N* = 182	*N* = 139	
1	24 (5.2%)	9 (6.3%)	13 (7.1%)	2 (1.4%)	0.72
2	21 (4.5%)	9 (6.3%)	2 (1.1%)	10 (7.2%)	
3	57 (12.3%)	15 (10.6%)	18 (9.9%)	24 (17.3%)	
4	94 (20.3%)	21 (14.8%)	44 (24.2%)	29 (20.9%)	
5	267 (57.7%)	88 (62.0%)	105 (57.7%)	74 (53.2%)	

* 1 expresses the lowest and 5 highest levels of anxiety; ** level of statistical significance of the Cochran–Mantel–Haenszel test.

**Table 2 ijerph-17-00985-t002:** The level of cancer-related anxiety delineated by the respondents. Descriptive statistics together with differences tests in subgroups defined by selected variables.

		Level of Anxiety (5-Point Likert Scale; 1 = Lowest, 5 = Highest)					
	Overall	1	2	3	4	5	*p*
	*N* = 463	*N* = 24	*N* = 21	*N* = 57	*N* = 94	*N* = 267	
**Age (Years)**							
Median (IQR)	33 (22–47)	42 (27.5–62)	33 (22–56)	35 (26–48)	29 (20.2–46)	33 (22–45)	0.1 ^k^
**Sex**							
Male	179 (38.7%)	9 (5.0%)	9 (5.0%)	19 (10.6%)	42 (23.5%)	100 (55.9%)	0.93 ^&^
Female	284 (61.3%)	15 (5.3%)	12 (4.2%)	38 (13.4%)	52 (18.3%)	167 (58.8%)	
**Place of Residence with the Number of Inhabitants ***							
≤100,000	218 (47.1%)	9 (4.1%)	8 (3.7%)	26 (11.9%)	45 (20.6%)	130 (59.6%)	0.2 ^&^
>100,000	245 (52.9%)	15 (6.1%)	13 (5.3%)	31 (12.7%)	49 (20.0%)	137 (55.9%)	
**Level of Education ****							
Lower	196 (42.3%)	9 (4.6%)	7 (3.6%)	24 (12.2%)	39 (19.9%)	117 (59.7%)	0.35 ^&^
Higher	267 (57.7%)	15 (5.6%)	14 (5.2%)	33 (12.4%)	55 (20.6%)	150 (56.2%)	
**Occupation**							
Nonmedical	388 (83.8%)	23 (5.9%)	16 (4.1%)	51 (13.1%)	78 (20.1%)	220 (56.7%)	0.17 ^&^
Medical	75 (16.2%)	1 (1.3%)	5 (6.7%)	6 (8.0%)	16 (21.3%)	47 (62.7%)	
**Positive Family History of Cancer**							
No	120 (25.9%)	15 (12.5%)	4 (3.3%)	19 (15.8%)	25 (20.8%)	57 (47.5%)	< 0.001 ^&^
Yes	343 (74.1%)	9 (2.6%)	17 (5.0%)	38 (11.1%)	69 (20.1%)	210 (61.2%)	
**Diagnosis of Cancer**							
No	413 (89.2%)	18 (4.4%)	19 (4.6%)	47 (11.4%)	87 (21.1%)	242 (58.6%)	0.02 ^&^
Yes	50 (10.8%)	6 (12.0%)	2 (4.0%)	10 (20.0%)	7 (14.0%)	25 (50.0%)	
**Treated for Cancer**							
No	415 (89.6%)	18 (4.3%)	19 (4.6%)	46 (11.1%)	90 (21.7%)	242 (58.3%)	0.017 ^&^
Yes	48 (10.4%)	6 (12.5%)	2 (4.2%)	11 (22.9%)	4 (8.3%)	25 (52.1%)	
**Self-Declaration of Cancer-Related Level of Knowledge**							
No	362 (78.2%)	17 (4.7%)	16 (4.4%)	42 (11.6%)	71 (19.6%)	216 (59.7%)	0.12 ^&^
Yes	101 (21.8%)	7 (6.9%)	5 (5.0%)	15 (14.9%)	23 (22.8%)	51 (50.5%)	
**Self-Declaration of Willingness to Improve the Level of Cancer-Related Knowledge**							
No	35 (7.6%)	2 (5.7%)	2 (5.7%)	7 (20.0%)	6 (17.1%)	18 (51.4%)	0.34 ^&^
Yes	428 (92.4%)	22 (5.1%)	19 (4.4%)	50 (11.7%)	88 (20.6%)	249 (58.2%)	

Note: IQR, interquartile range; k, Kruskal–Wallis test; &, Cochran–Mantel–Haenszel test (nonzero correlation); * Villages and cities having ≤100,000 inhabitants were combined to ≤100,000 category; ** Primary and secondary values were combined to form a single ‘low’ category, whereas both high school and high school medical were assigned to a ‘high’ category.

**Table 3 ijerph-17-00985-t003:** Differences in the level of anxiety associated with selected diseases: baseline and final assessment in intervention and control groups in terms of numbers and percentages (in brackets).

Level of Cancer-Related Anxiety *	Intervention			Control		
	Baseline	Final	*p* **	Baseline	Final	*p* **
1	15 (7.2%)	11 (5.3%)	0.03	9 (3.5%)	10 (3.9%)	0.6
2	12 (5.8%)	16 (7.7%)		9 (3.5%)	11 (4.3%)	
3	23 (11.1%)	34 (16.4%)		34 (13.3%)	26 (10.2%)	
4	41 (19.8%)	48 (23.2%)		53 (20.7%)	67 (26.2%)	
5	116 (56.0%)	98 (47.3%)		151 (59.0%)	142 (55.5%)	

* 1 expresses the lowest and 5 the highest level of anxiety; ** *p*, statistical significance in Wilcoxon test for paired variables.

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
