# Peer review of "Decreasing the Impact of Anxiety on Cancer Prevention through Online Intervention"

_ijerph, 2020, doi:10.3390/ijerph17030985_

Round 1
Reviewer 1 Report
This manuscript describes a statistical analysis on cancer-related anxiety based on the previous the survey study between the control and intervention groups. I have some statistical questions and comments as follows.
(1) The title of manuscript is “Decreasing the impact of anxiety on cancer prevention through online intervention”. However, in Line 90 – 82, the manuscript states that “No statistically significant differences were found in the levels of anxiety declared in the initial survey by the members of the control and 91 intervention groups (p=0.28 in Chi-square test).” The title is confusing and should be more specific/clear.
(2) "1118 volunteers were included in the study, 463 of whom with complete responses to the questionnaire were qualified for the final analysis." There were many missing values and there is no discussion on why and how so many missing values in this data set.
(3) There is limitation of p-value in Table 1 (from Chi-square test). The two variables are ordinal data, but Chi-square test ignores this information. You may use R package “vcdExtra” where a function of CMHtest (type= “cor”) can be used (taking into account the ordinal data).
(4) Again, CMHtest in R package “vcdExtra” can be used in Table 2 (choose the type = “cmeans” or type = “rmeans”)
(5) Wilcoxon test for paired data in Table 3 is too simple and not very good since the outcome variable = 1, 2, 3, 4, or 5. Ordinal logistic model is a better option. In addition, it can adjust for other confounding variables.
Reviewer 2 Report
Gajda et al., have reported that the internet-based intervention of knowledge provision would help alleviate the anxiety associated with people towards the methods of cancer diagnosis and treatment options. Authors have performed a much-needed study, especially with non-patient group of people. Authors have presented enough background information and systematically explained the methodologies used for the study and presented the results cohesively. I highly believe that the study would support to bring right changes in cancer care and recommend the acceptance of the paper however the authors should address the following concern with the citation.
In lines 179-182 authors mention a paper by Robb et al., that reported no significant change in the anxiety levels with increased awareness/knowledge to cancer. However, the authors Gajda et al., went on to say that they did not find any such relationship in their present study. In contrary, Table 3 of this paper lists anxiety level scale 2-4 in intervention (baseline) shows that online information has indeed increased the anxiety levels post intervention. Authors should discuss about it.
In line 77, page.no. 2, Reference 20 is cited for R software, but it appears to be a review article. Authors should make sure for R software citation.
Also, authors should talk about the accessibility of cancer related knowledge information available on the internet which may have been accessed by the participants. Since the internet has false negative articles that may increase the misconception in the readers that eventually increases the anxiety levels of the readers.
Also, authors should discuss why most of the participants failed to pass the test (only 463 completed the questioner out of 1118 participants) conducted after being exposed to it. It is >50% and appears that the content may have been difficult to understand by the participants.
Author Response
RESPONSES TO 2nd REVIEWER’S COMMENTS
“In line 77, page.no. 2, Reference 20 is cited for R software but its actually a review article. Authors should make sure for R software citation.”
AUTHORS’ RESPONSE:
We are very thankful for the time spent on paper evaluation and for pointing out this inaccuracy. References were corrected. Kindly see line 80 (line numbering refers to the revised version with enabled track changes option). Please note that subsequent references have been renumbered accordingly.

Reviewer 3 Report
This is a randomized, interventional study of an on-line educational program aimed to reduce self-declared cancer-related anxiety in order to promote preventive programs (the PORINA study, that demonstrated that an intervention of this kind may be useful in cancer prevention -Int J Environ Res Public Health. 2018 Jun 4;15(6)-).
The topic addressed is worthy of investigation. The Methodology is correctly designed and described with enough detail to understand the procedures adopted. The sample of participants is adequately described for the purposes of this work.
The study finds that i) respondents with low Cancer Knowledge Index values as well as positive family history of cancer declared the highest level of cancer-related anxiety, while lower levels were declared by those previously treated for cancer and ii) the intervention reduced the declared level of cancer-related anxiety. It concludes that the use of an intervention of this kind could reduce cancer-related anxiety and may lead to more frequent consent to undergo some of the medical procedures used to diagnose or treat cancer. The results, correctly discussed, are regarded as preliminary, and the limitations have been acknowledged in the discussion.
Please revise the English language in the manuscript and correct minor Grammar issues.
Author Response
RESPONSES TO 3rd REVIEWER’S COMMENTS
“Please revise the English language in the manuscript and correct minor Grammar issues.”
AUTHORS’ RESPONSE:
We are grateful to the Reviewer for a favorable assessment. The English of our manuscript was corrected with the help of native US English speaker (kindly see “Acknowledgments” section – lines 235-237). Line numbering refers to the revised version with enabled track changes option.
